# A Review of the Molecular Determinants of Therapeutic Response in Non-Small Cell Lung Cancer Brain Metastases

**DOI:** 10.3390/ijms25136961

**Published:** 2024-06-26

**Authors:** Catherine Boldig, Kimberly Boldig, Sepideh Mokhtari, Arnold B. Etame

**Affiliations:** 1Department of Neurology, University of South Florida, 2 Tampa General Circle, Tampa, FL 33606, USA; 2Department of Internal Medicine, University of Florida Jacksonville, 655 W. 8th St., Jacksonville, FL 32209, USA; kimberly.boldig@jax.ufl.edu; 3Moffitt Cancer Center, Department of Neuro-Oncology, 12902 USF Magnolia Drive, Tampa, FL 33612, USA; sepideh.mokhtari@moffitt.org (S.M.); arnold.etame@moffitt.org (A.B.E.)

**Keywords:** brain metastases, lung cancer, therapeutic targets, EGFR, precision therapy

## Abstract

Lung cancer is a leading cause of cancer-related morbidity and mortality worldwide. Metastases in the brain are a common hallmark of advanced stages of the disease, contributing to a dismal prognosis. Lung cancer can be broadly classified as either small cell lung cancer (SCLC) or non-small cell lung cancer (NSCLC). NSCLC represents the most predominant histology subtype of lung cancer, accounting for the majority of lung cancer cases. Recent advances in molecular genetics, coupled with innovations in small molecule drug discovery strategies, have facilitated both the molecular classification and precision targeting of NSCLC based on oncogenic driver mutations. Furthermore, these precision-based strategies have demonstrable efficacy across the blood–brain barrier, leading to positive outcomes in patients with brain metastases. This review provides an overview of the clinical features of lung cancer brain metastases, as well as the molecular mechanisms that drive NSCLC oncogenesis. We also explore how precision medicine-based strategies can be leveraged to improve NSCLC brain metastases.

## 1. Introduction

Lung cancer remains the leading cause of cancer-related deaths [1]. Although smoking has emerged as one of the greatest risk factors for lung cancer, lung cancer cases and deaths continue to rise despite a decrease in smoking prevalence [1]. Lung cancer is one of the most common malignancies to metastasize to the brain [2]. Lung cancer can be broadly classified as either small cell lung cancer (SCLC) or non-small cell lung cancer (NSCLC). NSCLC contributes to approximately 80% of lung cancer diagnoses and 70% of cases present with locally advanced or metastatic disease [3]. The prognosis of lung cancer brain metastasis remains grim with a median five-year survival rate of less than 5% [4]. Various molecular features and clinical characteristics of lung cancer have been associated with a greater likelihood of developing brain metastasis. Locally advanced NSCLC, age <60 years, non-squamous cell lung cancer, bulky mediastinal lymphadenopathy, genetic mutations of programmed cell death 1 (PD-1) protein, liver kinase B1 (LKB1) protein, *Kirsten rat sarcoma viral oncogene* (*KRAS*), *epidermal growth factor receptor* (*EGFR*), or rearrangements of *anaplastic lymphoma kinase* (*ALK*) portend a worse prognosis [5]. 

Brain metastases occur in a tumor microenvironment (TME). The TME is a diverse assortment of infiltrating and resident host cells interacting with and infiltrating into each other. Cells, soluble factors, and extracellular matrix are reprogrammed by tumor cells, impacting disease evolution and progression [6]. Lung cancer cells metastasize via epithelial-to-mesenchymal transition (EMT), a cellular process involving loss of cell–cell contact and acquisition of a mesenchymal phenotype [6,7]. These cells develop mutations that facilitate dissemination and survival in the circulatory system and subsequently, brain tissue. Certain molecular modifications are associated with a greater propensity for brain metastasis, such as mutations in *EGFR*, *ALK*, *KRAS*, or *c-ros oncogene 1* (*ROS1*) [7]. Tumor cells penetrate the blood–brain barrier (BBB) via the destruction of tight junctions and adherence to vascular endothelial cells. Tumor cells interact with astrocytes and immune cells via enzymes that subsequently allow metastatic lung cancer cells to proliferate. 

Treatment options for lung cancer brain metastases remain limited. Various actionable targets have changed the landscape of lung cancer treatment, specifically for NSCLC. Advances in NSCLC treatment have identified mutations and gene rearrangements, shifting the treatment paradigm from chemotherapy, radiation, and surgery to specific, targetable therapies. However, blood–brain barrier penetrance and acquired mutation-driven resistance remain critical obstacles to therapeutic efficacy. This review provides an overview of the clinical features of lung cancer brain metastases, as well as the molecular mechanisms that drive NSCLC oncogenesis. We also explore how precision medicine-based strategies can be leveraged to improve NSCLC brain metastases.

## 2. Histopathological Features

Lung cancer is characterized by histological features based on the composition of malignant cells. Variabilities in the cell of origin, as well as the degree of tumor differentiation, can present challenges in distinguishing NSCLC subtypes. NSCLC tumor subtypes include squamous cell, adenocarcinoma, large cell, neuroendocrine, and carcinomas with pleomorphic, sarcomatoid, or sarcomatous features [8,9]. 

Squamous cells, with less than 10% mucin, are classified as squamous carcinoma [8]. They have high levels of keratinization and well-formed classical bridges and are usually associated with the cell markers p40 and p63 [9]. Squamous carcinoma subtypes include keratinizing, nonkeratinizing, and basaloid, based on the presence of keratinization or if more than 50% of the tumor has basaloid features [9]. Basaloid features involve specific expression of mRNA, factors controlling cell cycle, transcription, chromatin, and splicing prevalent in germ cells and stem cells [9]. 

Adenocarcinoma can be acinar, papillary, lepidic, or micropapillary, characterized by more than 5 mm invasion and positive staining with pneumocyte markers such as thyroid transcription factor (TTF-1) or mucin [9,10]. Cells typically have septal widening with sclerosis and elastosis [10]. Large cell lung cancer (LCC) is not a true tumor entity but rather a term to describe undifferentiated cells that do not have squamous, neuroendocrine, or glandular features [11]. The cells are large and polygonal, arranged in groups with vesicular nuclei, prominent nucleoli, and moderate cytoplasm [11]. Carcinomas of pleomorphic, sarcomatoid, or sarcomatous elements contain spindle cell tumors composed of epithelial origin and pleomorphic carcinoma [8]. Spindle cell tumors are composed of elongated sarcoma-appearing cells, while pleomorphic carcinoma is composed of squamous or adenocarcinoma with at least 10% spindle or giant cells [8]. Subtyping lung cancer based on histopathological features allows further identification of molecular features to provide the most targeted treatment.

## 3. Radiographic Presentation of NSCLC Brain Metastases

Magnetic resonance imaging (MRI) is the gold standard for detecting brain metastasis. Standard sequences include pre- and post-contrast 3D T1-weighted (T1W) and T2-weighted (T2W) with fluid-attenuated inversion recovery (FLAIR) [12]. Metastases are usually iso- or hypointense in T1W imaging and have variable signal intensities in T2W depending on cystic versus solid content [12]. Brain metastases cause surrounding vasogenic edema, affecting the white matter, and are typically found at the grey–white junction or watershed zone due to hematogenous spread [12]. This vasogenic edema is well appreciated in T2 FLAIR sequences. A common imaging feature is ring enhancement due to central necrosis or cystic contents [12]. Diffusion-weighted MRI sequences can be used to distinguish ring-enhancing metastases from abscesses. The latter demonstrates restricted diffusion while the former might not. Moreover, differentiating recurrent brain metastasis from radiation-related changes on MRI can be challenging [13]. MRI perfusion, generally, demonstrates a significant increase in blood flow in recurrent tumors compared to radiation-related changes. Further MRI perfusion could potentially differentiate primary brain tumors from brain metastases because primary tumors typically have higher relative cerebral blood flow [12]. MRI spectroscopy can also be useful in distinguishing non-small cell lung cancer from other metastases using the standardized (cho)/creatine (cr)-ratio, which is typically greater than 2 [12]. Lung cancer is typically associated with multiple sites of brain metastases, and the location may vary depending on the type; however, overall, the highest incidence occurs in the cerebellum or frontal lobe [14].

## 4. Standard Treatment Modalities for NSCLC

The treatment of lung cancer has evolved in recent years. Previously, chemotherapy was the cornerstone of treatment, and this has evolved to targeted approaches such as immunotherapy. Concurrently, there has been significant progress in the treatment of lung cancer brain metastases. Surgical resection is performed for primary lung site disease in 28% of patients with squamous cell carcinoma, 37.6% with adenocarcinoma, and 34.4% with other histological types [8]. Surgical resection and platinum-based chemotherapy are recommended for early stages (I, II, and IIIA) [9]. Most patients with stage III are not surgical candidates and are treated with chemoradiation and immunotherapy [9]. Platinum-based chemotherapy has shown an objective response rate of 21–50% for patients with NSCLC with brain metastasis [10,11,12,13,14,15,16,17,18,19]. Adjuvant cisplatin-based chemotherapy allows a 5.4–11% five-year survival benefit compared to observation in patients with resected stage I to III NSCLC [20,21,22]. Neo-adjuvant chemotherapy has shown a similar survival benefit of 5–6% at five years [23]. The standard of care for patients with advanced stage (IV) NSCLC without a targetable mutation is immunotherapy monotherapy or in conjunction with chemotherapy [9].

## 5. Molecular Features of NSCLC

The molecular features (Figure 1) of NSCLC vary depending on the subtype. Adenocarcinoma has been shown to alter the receptor tyrosine kinase (RTK) of the *RAS*/*rapidly accelerated fibrosarcoma* (*RAF*) signaling pathway in approximately 75% of cases [24]. Most often mutations of *KRAS*, *EGFR*, and *v-raf murine sarcoma viral oncogene homolog B* (*BRAF*) drive the pathway into tumorigenesis [24]. Less commonly, *mesenchymal–epithelial transition factor* (*MET*), *human epidermal growth factor receptor 2* (*HER2*), *ROS1*, *ALK*, and *rearranged during transfection* (*RET*) are responsible for cellular transformation [24]. Messenger ribonucleic acid (mRNA) profiling has found three different transcriptional subtypes of adenocarcinoma. The terminal respiratory unit (TRU) subtype is often affected by *EGFR* mutations and kinase fusions, the proximal inflammatory (PI) subtype is characterized by *neurofibromatosis 1* (*NF1*) and *tumor protein P53* (*TP53*), and the proximal proliferative (PP) subtype frequently has mutations in *KRAS* and inactivation of *serine/threonine kinase 11* (*STK11*) [24]. Genetic mutations significant to squamous cell carcinoma include *TP53*, *human leukocyte antigens-A* (*HLA-A*), *kelch-like ECH-associated protein 1* (*KEAP1*), and *nuclear factor* (*erythroid-derived 2*)*-like 2* (*NFE2L2*). *TP53* is the most common mutation, occurring in 90% of cases [24]. 

Numerous molecular modifications to cancer cells occur to facilitate metastasis. Epithelial–mesenchymal transition (EMT) occurs through growth factor receptor tyrosine kinases and cellular signaling pathways and allows cell differentiation and proliferation [5]. Cell adhesion molecules such as E-cadherin are decreased in brain metastases [5,25]. Upregulated mesenchymal markers such as N-cadherin and vimentin contribute to cell migration during metastasis [5,26,27]. Cellular processes occur to allow cancer cell perpetuation through the BBB. The BBB becomes hyperpermeable, allowing brain metastasis [5,28]. The process requires disrupted immune regulations that allow the perpetuation of the metastatic disease within the BBB. Often, brain metastases have fewer T-cell infiltrates and increased microglia and monocytes, decreasing cell-mediated responses [29]. Brain metastasis from lung cancer appears to activate signaling pathways via hypomethylation [30]. *Lysine* (*K*)*-specific methyltransferase 2C* (*KMT2C*) is mutated in several cancers and is thought to be a tumor suppressor in small cell lung cancer. This pathway leads to hypomethylation of deoxyribonucleic acid methyltransferase 3A (DNMT3A), which activates pro-metastatic genes [31].

## 6. Prognostic Oncogenic Drivers in NSCLC

The molecular characteristics (Figure 1) of lung cancer affect prognosis based on targeted treatment options and their propensity to metastasize. Oncogenic mutations in NSCLC largely impact treatment and survival. Tyrosine kinase receptors consist of a family that includes EGFR, HER1 (ERBB1), HER2 (ERBB2), HER3 (ERBB3), and HER4 (ERBB4) [32]. These receptors work through ligand binding and resultant dimerization. However, when a gain-of-function mutation occurs in the tyrosine kinase domain, the receptors work irrespective of ligand binding [32]. Identifying *EGFR*-activating mutations has shifted the paradigm in the treatment of NSCLC. 

Additionally, *ALK*, *ROS1*, *neurotrophic tyrosine receptor kinase* (*NTRK*), *RET*, and *neuregulin1* (*NRG1*) are associated with gene rearrangements and mutations driving new systemic therapies [7,33]. Genetic alterations allow cellular cascades through the phosphatidylinositol-3 kinase (P13k)/Akt/mammalian target of rapamycin (mTOR), Janus kinase (JAK)/signal transducer and activator of transcription (STAT), protein kinase C (PKC), and mitogen-activated protein kinase (MAPK) pathways to cause cellular proliferation and tumorigenesis, as seen in Figure 1 [7,32]. The key oncogenic drivers and their associated frequencies in NSCLC are listed in Table 1 [3,34,35,36,37,38,39,40,41,42,43,44].

Although brain metastases are quite common in NSCLC, there appears to be an increased propensity for brain metastases in NSCLC patients with oncogenic drivers. Alterations of the *EGFR*, *ALK*, *KRAS*, and *ROS1* genes are associated with a higher incidence of brain metastasis [7,45,46,47,48,49,50,51,52,53,54]. Studies have reported a 19–50% increased incidence of brain metastasis in patients with these gene fusions, rearrangements, or mutations, as seen in Table 2 [7,45,46,47,48,49,50,51,52,53,54,55,56,57,58,59]. Tyrosine kinase inhibitors (TKIs) have become the first-line treatment option for patients with advanced disease with these mutations or rearrangements [7]. Given the robust intracranial response from several TKIs, there is a paradigm shift toward employing targeted therapy as first-line therapy for patients with asymptomatic brain metastases.

## 7. Precision Medicine Strategies for NSCLC Brain Metastases

A longstanding challenge in neuro-oncology is the limited penetrance of systemic chemotherapy into the brain. Hence, despite excellent systemic control in several instances, patients have succumbed to intracranial disease progression. The BBB is a critical impediment to the delivery of therapeutics into the central nervous system. Although the BBB is permeable to cancer cells, the permeability of therapeutics remains low, limiting the utility of standard chemotherapy options in patients with brain metastases [7]. Treatment advances have led to the development of tyrosine kinase inhibitors (TKIs) with increased permeability of the BBB, offering promise for the treatment of brain metastasis [60,64,75,76,77]. Table 2 summarizes the advances in precision medicine strategies for brain metastases with key druggable oncogenic drivers and the associated overall intracranial response rates [60,61,63,64,65,66,67,68,69,70,71,72,73,74].

### 7.1. Precision Medicine Strategies for EGFR-Mutant NSCLC

In light of the higher incidence of brain metastases in patients with EGFR mutations, targeted therapy with brain-penetrant TKIs offers the greatest probability of therapeutic success [78]. It is estimated that up to 20–30% of patients with EGFR-mutated NSCLC develop brain metastases (Table 2) [45,46,47,48]. EGFR TKIs have shown better outcomes in the treatment of NSCLC with EGFR mutations compared to chemotherapy [79]. The first-generation EGFR TKIs erlotinib and gefitinib have shown improved progression-free survival but little impact on overall survival [79,80,81,82,83,84,85]. The second-generation EGFR TKI afatinib has shown improved progression-free survival and overall survival in a subset of patients with 19 exon deletion EGFR mutations compared to chemotherapy [86,87,88]. Dacomitinib, also a second-generation EGFR TKI, has shown improved progression-free survival compared to erlotinib [89]. The effects of first- and second-generation EGFR TKIs may be limited by patient-acquired or intrinsic resistance [79]. 

Third-generation EGFR TKIs were developed to target these cellular modifications, specifically the T790M mutation, which is present in 49–63% of NSCLC cases [79,90,91,92]. Osimertinib, rociletinib, olmutinib, and AC0010 have shown improved progression-free survival and objective response rates in patients with T790 mutations [93,94,95,96,97,98]. Furthermore, in patients with brain metastases, osimertinib has demonstrated an intracranial overall response rate of 66% (Table 2) [60]. Third-generation EGFR TKIs were designed to bind to the ATP binding site of the EGFR receptor. However, resistance is again acquired through a C797S mutation at the ATP binding site [99]. 

Subsequently, fourth-generation EGFR TKIs have been developed to overcome this mutation-driven resistance [99,100]. EAI045 is a fourth-generation allosteric inhibitor of EGFR that binds at a site away from the ATP binding site. EAI045 is most efficacious when combined with cetuximab, which prevents dimerization of EGFR [99]. EAI045, in combination with cetuximab, was found to be effective in NSCLC mouse models driven by L858R/T790M EGFR and by L858R/T790M/C797S EGFR [99,101,102]. CH7233163 is another promising allosteric inhibitor that is highly selective and potent against EGFR-Del19/T790M/C797S, L858R/T790M/C797S, L858R/T790M, Del19/T790M, Del19, and L858R in preclinical models [103]. Lastly, JBJ-04-125-02 is an allosteric inhibitor that is effective against L858R/T790M/C797S-mutated NSCLC and that synergizes with osimertinib to delay the acquisition of resistance [104]. The potential of fourth-generation allosteric inhibitors of EGFR to overcome osimertinib resistance appears very promising, especially if these new agents are highly brain-penetrant. Moreover, EGFR is a molecular target highly associated with NSCLC brain metastases for which precision medicine strategies with targeted therapies have improved intracranial outcomes [105]. 

### 7.2. Precision Medicine Strategies for RET-Rearranged NSCLC

Multikinase inhibitors were the first tested treatment in RET-rearranged NSCLC; however, their efficacy is limited [106,107]. They demonstrate response rates of 16–47%, with 53–73% of patients requiring dose reductions due to drug toxicity [108]. Newer RET TKIs have shown increased efficacy with less toxicity. Selpercatinib (LOXO-292) has shown an 85% objective response rate in the LIBRETTO-001 study in patients with central nervous system (CNS) metastases (Table 2) [61]. Selpercatinib has also shown radiologic disease control and symptom resolution in a patient with NSCLC with leptomeningeal spread previously treated with multikinase inhibitors and stereotactic radiosurgery treatments [109]. Pralsetinib (ARROW) showed a 51% overall response rate in patients with NSCLC and CNS involvement with previous platinum-based treatment and a 70% response rate in treatment-naïve patients [108]. All patients with measurable intracranial metastases showed shrinkage and 11% demonstrated a complete response [108]. Although RET fusions are present only in 1–2% of NSCLC cases (Table 1), there is a 50% lifetime prevalence of brain metastases (Table 2) [40,61]. Promising precision agents such as selpercatinib (LOXO-292) undoubtedly have a significant impact on outcomes for NSCLC brain metastasis patients with RET fusions. 

### 7.3. Precision Medicine Strategies for ALK-Rearranged NSCLC

ALK translocations have been associated with a higher percentage of brain metastasis, occurring in 23–34% of patients with NSCLC (Table 2) [57,62,110]. Crizotinib, a first-generation ALK, MET, and ROS1 tyrosine kinase inhibitor (TKI) has demonstrated superior progression-free survival and objective response rates compared to conventional chemotherapy [52,111,112,113]. Moreover, crizotinib has been reported to have good intracranial disease control [62]. The second-generation ALK TKIs, alectinib and ceritinib, appear to be more efficacious with respect to intracranial disease control [114,115,116]. For instance, alectinib has shown an 81% CNS response rate (Table 2) compared to 50% with crizotinib, demonstrating superior CNS penetrance [114]. In addition, in patients with symptomatic brain metastasis or leptomeningeal disease, alectinib has led to improvement in clinical symptoms [117]. Similar to first-generation TKIs, resistance often develops to second-generation TKIs, which has led to the development of third-generation agents. The third-generation lorlatinib has demonstrable systemic and intracranial efficacy, both in treatment-naïve patients and those with treatment-acquired resistance mutations, to early-generation ALK inhibitors [118,119,120]. In ALK-rearranged NSCLC, the intracranial response rate of lorlatinib is 82% (Table 2) [66]. Collectively, in a systematic review of 21 studies of ALK inhibitors, including crizotinib, ceritinib, alectinib, and brigatinib, the pooled intracranial disease control rate was 70.3% and 79% in first-line treatment and pre-treated patients, respectively [121]. 

### 7.4. Precision Medicine Strategies for ROS1-Rearranged NSCLC

Although crizotinib targets ROS1, its utility in ROS1-positive NSCLC is limited by CNS activity and acquired resistance [122]. Approximately 30% of patients on crizotinib develop CNS metastasis during treatment, making the CNS one of the most common sites of therapeutic failure [122]. A combination of CNS penetrance limitations and an aggressive ROS1-mediated cancer phenotype could be contributory factors to CNS failure. Lorlatinib, an ALK/ROS1 inhibitor, has demonstrated a 53–67% objective response rate in crizotinib-pretreated and naïve patients [123]. Lorlatinib has also shown a 64% intracranial response rate in patients with ROS1 NSCLC (Table 2) [67]. Repotrectinib, a ROS1/NTRK/ALK inhibitor, showed intracranial activity in the Trident 1 study, as well as a 100% intracranial response rate (Table 2) [69,124]. DS-6051b, a ROS1/pan-NTRK inhibitor, has demonstrated a 75% objective response rate and 100% disease control rate in patients with NSCLC and brain metastases [125]. 

### 7.5. Precision Medicine Strategies for KRAS-Mutant NSCLC

KRAS are one of the most common and studied mutations in NSCLC, yet this driver has the most limited therapeutic targets. KRAS mutations affect up to 30% of NSCLC cases [41,126], and CNS metastases portend a worse prognosis in KRAS-mutated NSCLC patients compared to non-KRAS-mutated patients [71,127]. A significant challenge in targeting KRAS is its high affinity for GTP as opposed to ATP like other tyrosine kinases. However, recent advances in small molecule discoveries have led to the development of inhibitors that target the cysteine site of KRAS–GDP [126]. Sotorasib and adagrasib are KRAS inhibitors that have shown 88% and 42% overall intracranial response rates, respectively, in clinical trials (Table 2) [70,71,127]. In this regard, sotorasib appears to be a promising agent for intracranial control of KRAS-mutated NSCLC.

### 7.6. Precision Medicine Strategies for MET-Altered NSCLC

MET exon 14 skipping mutations (METex14) and MET amplifications impact approximately 1–6% of NSCLC cases [72]. Brain metastases have a reported incidence of 11–23% of patients with NSCLC MET exon 14 mutations (Table 2) [58]. Several MET pathway inhibitors have been studied and assessed for intracranial activity. Capmatinib, a MET receptor inhibitor, showed an intracranial response in 7 of 13 patients with complete resolution in 4 patients [72]. Studies have shown tepotinib to have a 55% response rate in 11 patients with brain metastasis and an intracranial response rate of 71% (Table 2) [73,128]. Savolitinib has demonstrated an objective response rate of 49.2% and adequate control of brain metastasis [129]. 

### 7.7. Precision Medicine Strategies for BRAF/MAPK NSCLC

MAPK pathway is an important cellular pathway with mutations leading to tumorigenesis. It comprises kinases such as RAS, RAF, MEK, and ERK. The BRAF proto-oncogene contributes to 2–5% of NSCLC mutations (Table 1), of which BRAF V600E is the most common [3,37]. Clinical trials have shown MEK inhibitors to lack efficacy and promise as a monotherapy in the treatment of NSCLC [3,130,131]. Alternatively, BRAF inhibitors have been found to be an effective monotherapy. Vemurafenib has shown an effective response in BRAF-mutated NSCLC, activity against brain metastases, and meningeal carcinomatosis [132,133,134]. However, BRAF inhibitor therapy has been found to be limited by the development of resistance. Pairing BRAF and MEK inhibitors has demonstrated increasing effectiveness for MEK inhibitors and limits the development of BRAF inhibitor resistance [134]. 

Dabrafenib is a BRAF kinase inhibitor selective for the BRAF V600E mutation [135]. When trametinib, a MEK inhibitor, was paired with dabrafenib, results were promising. This combination therapy showed an objective response rate of 68.4% in pretreated patients; 5% had a complete response, and 63% had a partial response [135]. Further assessment of the impact on CNS metastasis is required. A multicenter study reported that one patient with brain metastasis showed resolution upon imaging on follow-up, but four patients developed brain metastases during treatment [136]. However, a CNS response has been seen with dabrafenib–trametinib in other cancers with CNS involvement such as melanoma and high-grade glioma [137,138]. The pan-RAF inhibitors CCT196969 and CCT241161 are undergoing early investigations, appearing to suppress mutant BRAF cells without activating the downstream MAPK pathway [134]. 

### 7.8. Precision Medicine Strategies for NTRK-Rearranged NSCLC

NTRK gene fusion and amplification is a targetable oncogenic driver in NTRK-positive NSCLC patients for which several inhibitors have been developed. The first-generation NTRK inhibitors, larotrectinib and entrectinib, have been reported to have overall response rates of 57–80% in clinical trials [139,140,141,142,143,144]. Both agents have demonstrated overall CNS response rates of between 50% and 75% [139]. Given the longstanding history of resistance acquisition to early TKIs, there has been an emphasis on developing advanced next-generation TRK inhibitors [139]. Selitrectinib, a selective TRK inhibitor; repotrectinib, an ALK/ROS1/pan-TRK inhibitor; and taletrectinib, an NTRK and ROS1 inhibitor, are undergoing evaluation of NTRK-positive NSCLC, with the need to further understand the impact on CNS metastases [139]. 

### 7.9. Precision Medicine Strategies for HER2-Rearranged NSCLC

Alterations in HER2 transpire through three mechanisms to cause tumorigenesis in NSCLC. Gene mutations, gene amplifications, and protein overexpressions trigger cellular cascades and cell proliferation [145]. CNS involvement has been reported in up to 47% of HER2-mutant NSCLC (Table 2) [59,145]. Initial studies of the nonselective HER2 inhibitors afatinib, dacomitinib, and neratinib showed variable effectiveness in treatment [146]. Selective HER2 inhibitors have shown more promise in treatment. Poziotinib is an EGFR/HER2 inhibitor that showed, in the ZENITH20 trial, a 28–39% objective response rate, a 70–73% disease control rate, and a 74–80% tumor reduction rate in pretreated and treatment-naïve NSCLC patients with HER2 20 exon mutations [147,148]. In ZENITH20 Cohorts 1–3, 80% of patients had stable CNS disease, 3/36 patients had intracranial complete responses, with an 89% disease control rate (Table 2) [74]. Trastuzumab is a monoclonal antibody that binds to the HER2 receptor and prevents its dimerization. It has shown limited impact in the treatment of HER2-positive NSCLC [145]. 

### 7.10. Precision Medicine Strategies for NRG1-Rearranged NSCLC

Additional molecular targets in the treatment of NSCLC are under ongoing investigation. The incidence of NRG1 fusions in NSCLC is approximately 0.1–0.3% (Table 1) with a brain metastasis incidence of 15% [7,43,44]. NRG1 fusions are associated with binding HER3, causing HER2–HER3 heterodimerization. Because of this association, HER2/3-targeted therapies are being studied for their effectiveness in NRG1-targeted therapies [7]. FGFR fusions, phosphatidylinositol-4,5-bisphosphate 3-kinase catalytic subunit alpha (PI3KCA) mutations, and discoidin domain receptor tyrosine kinase 2 (DDR2) mutations are additional rare genetic alterations associated with NSCLC requiring further investigation in terms of their relationship with CNS metastases [111].

### 7.11. Other Precision Medicine Strategies for NSCLC

Another novel treatment approach consists of antibody–drug conjugates (ADCs). These emerging agents utilize the monoclonal antibody binding capabilities and chemotherapy cytotoxicity to target HER2-altered NSCLC tumor cells [145]. Trastuzumab emtansine (TDM1) has been shown as an effective treatment in HER2-altered NSCLC [145]. Trastuzumab deruxtecan (DS-8201a) and trastuzumab duocarmazine (SYD985) were developed to reduce TDM1 resistance and increase efficacy. DS-8201a has shown antitumor activity with an objective response rate of 55–73% [149,150,151]. Further research is required to identify the impact on CNS metastases. 

Finally, in addition to single-agent precision medicine approaches, combinational therapeutic strategies have been devised for the treatment of NSCLC brain metastases. Potential synergies from multimodal targeting should enhance therapeutic efficacy and ameliorate therapeutic resistance. Such strategies could entail combinations of any or all of the following: Standard chemotherapy, precision medicine targeting, immunotherapy, and radiotherapy. Further exploration of combinatorial approaches is beyond the scope of this review.

## 8. Discussion

Lung cancer accounts for the vast majority of brain metastases and remains a leading cause of cancer-related death with NSCLC as the most predominant histology subtype. Because 20–40% of patients with NSCLC have brain metastasis, understanding treatment efficacy and toxicities is crucial for the treatment of these patients [152]. Cisplatin-based chemotherapy and surgical resection have been the cornerstone of NSCLC treatment. The Lung Adjuvant Cisplatin Evaluation (LACE) meta-analysis demonstrated a 5.4% absolute survival benefit at five years, and the hazard ratio of death was 0.89 with cisplatin-based adjuvant chemotherapy in NSCLC treatment [22]. The combined use of carboplatin and paclitaxel offered a 20% intracranial response rate in patients with NSCLC brain metastasis [152]. In combination with whole-brain radiotherapy, cisplatin or pemetrexed exhibited 68.3% and 34.1% intracranial response rates, respectively [152]. The intracranial response rates of precision agents against oncogenic drivers in NSCLC with brain metastasis range from 42 to 100%. These targeted therapies have emerged as efficacious treatment options in advanced NSCLC cases thanks to recent advances in molecular genetics coupled with innovations in small molecule drug discovery strategies.

The targeted treatments have shown intracranial response rates meaningful to the treatment of NSCLC patients with brain metastasis. These targeted therapies impact NSCLC treatment beyond use in patients with brain metastasis. These same medications have proven greater progression-free survival (PFS), disease control rates, objective response rates (ORR), and overall survival (OS) when studied in patients with NSCLC. Osimertinib, a third-generation EGFR-TKI, demonstrated in the FLAURA trial, a significant prolonging of PFS and OS relative to first-generation EGFR-TKIs in patients with previously untreated NSCLC [153]. The Libertto-001 was a phase I/II trial that showed an objective response rate of 84% and a 6% complete response rate in treatment-naive patients treated with selpercatinib [61]. The ALTA-1L trial was a phase 3 trial that showed improved PFS of briatinib over crizotinib in ALK inhibitor-naïve ALK-positive NSCLC [154]. The ALEX trial, a phase 3 trial, showed improved PFS of alectinib over crizotinib in untreated ALK-positive NSCLC [114]. Lorlatinib showed significantly longer PFS over crizotinib in treatment-naïve ALK-positive NSCLC in the CROWN study [66]. Lorlatinib, a ROS/ALK inhibitor, has also shown an objective response in patients with advanced ROS1-positive NSCLC [67]. Analysis of ALKA-372-001, STARTRK-1, and STARTRK-2 trials demonstrated prolonged survival with the use of entrectinib in ROS1 TKI-naïve ROS1 fusion-positive NSCLC [155]. The TRIDENT-1 trial has shown high response rates and durable responses with repotrectinib in ROS-positive NSCLC [156]. The CodeBreaK100 trial demonstrated an objective response of 37% and a complete response rate of 3.2% with sotorasib in patients with previously treated KRAS p.G12C-mutated NSCLC [157]. Participants enrolled in the KRYSTAL-1 study with KRAS^G12C^-mutated NSCLC had a disease control rate of 90% [71]. The GEOMETRY mono 1 study assessed capmatinib in patients with MET-amplified NSCLC and showed an overall response rate (complete or partial) in 29% of previously treated patients and 40% of patients not previously treated [72]. The VISION trial found tepotinib had a systemic efficacy in patients with METex14 skipping mutation NSCLC [128]. The ZENITH20 trial reports a 28% ORR with ponziotinib in patients with HER2 exon 20 mutations with advanced or metastatic NSCLC [148]. These targeted therapies have changed NSCLC treatment and have improved survival in patients with dismal prognoses. However, awareness of the shortcomings of these therapies is important to adapt treatment regimens and progress treatment options. 

The precision agents have limitations that include BBB penetrance, acquired resistance, and intratumoral heterogenicity. Early-generation precision agents often lack BBB penetrance, limiting their efficacy in brain metastasis. Later-generation agents are associated with weaker efflux transporter substrates in the brain, resulting in greater penetrance [158]. The NSCLC cells also develop resistance to the precision agents through mutations, gene amplifications, rearrangements, or activating downstream mechanisms to tumorigenesis pathways [159]. Intratumor heterogeneity enables cancer cells to survive treatment and propagate resistant phenotypes, maintain distinct genetic alterations within the same tumor, and allow cell-to-cell variation of protein expression [159]. Newer generation agents have been developed to overcome these mechanisms of resistance. To maintain targeted therapy for patients, identifying resistance mechanisms requires repeating tissue sampling via invasive measures, identifying circulating cell-free DNA in the blood of cerebrospinal fluid (CSF), or single-cell profiling [159]. 

Clinical intracranial response data with brain-penetrant agents targeting oncogenic drivers in NSCLC are very promising. However, therapeutic resistance remains inevitable, thereby necessitating the proactive development of next-generation therapeutics informed through knowledge from acquired resistance mechanisms. Immunotherapy has entered the treatment paradigm of NSCLC and has shown improvement in ORR [160]. Combination therapies targeting oncogenic drivers, TKIs, and immune checkpoint inhibitors may allow further targeted treatment regimens and precision therapy for NSCLC patients. It is, nonetheless, apparent that precision medicine is shifting the paradigm of treatment for brain metastases.

## Figures and Tables

**Figure 1 ijms-25-06961-f001:**
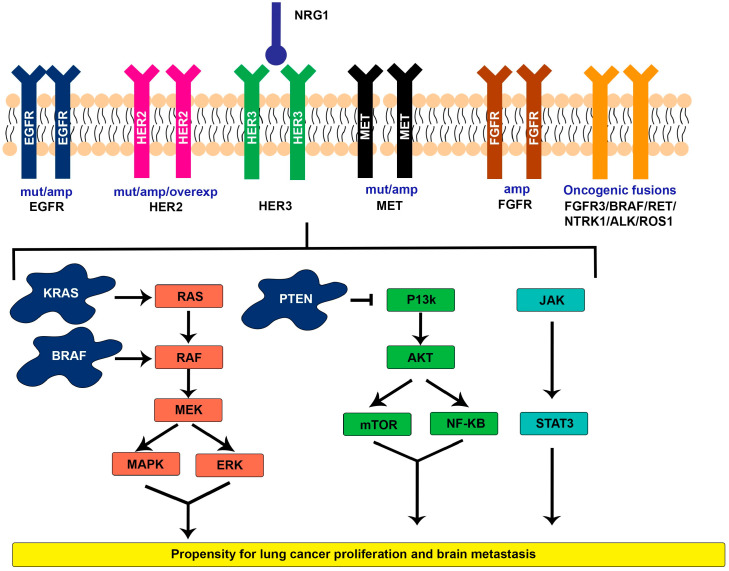
Molecular determinants and cellular cascades that increase the propensity for lung cancer proliferation and brain metastasis. *anaplastic lymphoma kinase* (*ALK*), amplification (amp), *v-raf murine sarcoma viral oncogene homolog B* (*BRAF*), *epidermal growth factor receptor* (*EGFR*), extracellular signal-related kinases (ERK), *fibroblast growth factor receptor* (*FGFR*), *human epidermal growth factor receptor* (*HER*), Janus kinase (JAK), *Kristen rat sarcoma viral oncogene homolog* (*KRAS*), mitogen-activated protein kinase (MAPK), mitogen-activated extracellular signal-regulated kinase (MEK), *mesenchymal*–*epithelial transition factor* (*MET*), mammalian target of rapamycin (mTOR), mutation (mut), nuclear factor-ĸB (NF-KB), neuregulin 1 (NRG1), *neurotrophic tyrosine receptor kinase* (*NTRK*), overexpression (overexp), protein kinase B (PKB/AKT), *phosphatase and tensin homolog deleted on chromosome 10* (*PTEN*), phosphatidylinositol-3 kinase (P13k), rapidly accelerated fibrosarcoma (RAF), RAS (rat sarcoma), *rearranged during transfection* (*RET*), *c-ros oncogene 1* (*ROS1*), signal transducer and activator of transcription (STAT).

**Table 1 ijms-25-06961-t001:** Key oncogenic drivers in NSCLC.

Oncogenic Driver	Precision Agent	Incidence (Ref.)
EGFR mutations	Erlotinib, gefitinib, dacomitinib, osimertinib	14% [34]
ALK rearrangements	Crizotinib, ceritinib, brigatinib, alectinib, lorlatinib	4–5% [35]
ROS1 rearrangements	Crizotinib, lorlatinib, entrectinib, repotrectinib	1–2% [36]
BRAF mutations	Dabrafenib, trametinib	2–5% [3,37]
MET mutations	Crizotinib, cabozantinib, capmatinib, tepotinib	3–4% [38]
NTRK fusions	Entrectinib, larotrectinib	0.1–1% [39]
RET fusions	Selpercatinib, pralsetinib	1–2% [40]
KRAS mutations	Sotorasib, adagrasib	20–25% [41]
HER2 mutations	Trastuzumab, deruxtecan	1–3% [42]
NRG1 rearrangements	Afatinib	0.1–0.3% [43,44]

*naplastic lymphoma kinase (ALK), v-raf murine sarcoma viral oncogene homolog B (BRAF), epidermal growth factor receptor (EGFR), human epidermal growth factor receptor 2 (HER2), Kristen rat sarcoma viral oncogene homolog (KRAS), mesenchymal–epithelial transition factor (MET), neuregulin 1 (NRG1), neurotrophic tyrosine receptor kinase (NTRK), rearranged during transfection (RET), c-ros oncogene 1 (ROS1)*.

**Table 2 ijms-25-06961-t002:** Intracranial response rates of precision agents in NSCLC brain metastases.

Oncogenic Driver	Brain Metastasis Rates	Precision Agent	Intracranial Response Rates
EGFR	20–30% [45,46,47,48]	Osimertinib	66% [60]
RET	* 50% [54]	Selpercatinib (LOXO-292)	85% [61]
ALK	23–34% [49,50,51,52,57,62]	Brigatinib	78% [63]
		Alectinib	81% [64,65]
		Lorlatinib	82% [66]
ROS1	19% [53]	Lorlatinib,	64% [67]
		Entrectinib	79% [68]
		Repotrectinib	100% [69]
KRAS	29% [55]	Sotorasib	88% [70]
		Adagrasib	42% [71]
MET	11–23% [58]	Capmatinib	50% [72]
		Tepotinib	71% [73]
HER2	47% [59]	Poziotinib	89% [74]

* lifetime prevalence. *anaplastic lymphoma kinase (ALK), epidermal growth factor receptor (EGFR), human epidermal growth factor receptor 2 (HER2), Kristen rat sarcoma viral oncogene homolog (KRAS), mesenchymal–epithelial transition factor (MET), rearranged during transfection (RET), c-ros oncogene 1 (ROS1)*.

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
