# Peer review of "A Review of the Molecular Determinants of Therapeutic Response in Non-Small Cell Lung Cancer Brain Metastases"

_ijms, 2024, doi:10.3390/ijms25136961_

Round 1

Reviewer 1 Report

Comments and Suggestions for Authors

Dear colleagues, 

After carefully reviewing the manuscript, I have the following comments and suggestions:

1. The review provides a comprehensive overview of the molecular determinants of therapeutic response in NSCLC brain metastases. However, to strengthen the analysis, it would be beneficial to include more in-depth discussion of key results from seminal clinical trials mentioned. Direct comparisons of relevant trials would help evaluate efficacies of different targeted therapies.

2. The tables are clearly organized and effectively summarize important oncogenic drivers and associated precision medicine strategies. To improve accessibility, it may help to include a brief glossary defining relevant molecular and clinical terminology for non-expert readers. 

3. The discussion of precision medicine strategies could be strengthened by a more nuanced analysis of limitations and challenges, such as varying degrees of blood-brain barrier penetrance, acquired resistance mutations, intratumoral heterogeneity, etc. 

In terms of the language, the writing is generally clear and concise with appropriate citation of primary sources. Minor areas for improvement include consistent use of medical terminology and avoiding unnecessary wordiness at times. 

A major strength is the comprehensive yet succinct overview of this vast topic. A potential limitation is that discussing every aspect briefly may compromise depth for some concepts. 

In closing, I would like to commend the authors for their impressive scholarly effort in aggregating and synthesizing this specialized knowledge base. Continued rigorous work in this important field has potential to greatly improve patient outcomes. I have full confidence that with further refinement, this review will make a valuable contribution to advancing precision cancer care for brain metastases.

Comments on the Quality of English Language

Dear colleagues, 

After carefully reviewing the manuscript, I have the following comments and suggestions:

1. The review provides a comprehensive overview of the molecular determinants of therapeutic response in NSCLC brain metastases. However, to strengthen the analysis, it would be beneficial to include more in-depth discussion of key results from seminal clinical trials mentioned. Direct comparisons of relevant trials would help evaluate efficacies of different targeted therapies.

2. The tables are clearly organized and effectively summarize important oncogenic drivers and associated precision medicine strategies. To improve accessibility, it may help to include a brief glossary defining relevant molecular and clinical terminology for non-expert readers. 

3. The discussion of precision medicine strategies could be strengthened by a more nuanced analysis of limitations and challenges, such as varying degrees of blood-brain barrier penetrance, acquired resistance mutations, intratumoral heterogeneity, etc. 

In terms of the language, the writing is generally clear and concise with appropriate citation of primary sources. Minor areas for improvement include consistent use of medical terminology and avoiding unnecessary wordiness at times. 

A major strength is the comprehensive yet succinct overview of this vast topic. A potential limitation is that discussing every aspect briefly may compromise depth for some concepts. 

In closing, I would like to commend the authors for their impressive scholarly effort in aggregating and synthesizing this specialized knowledge base. Continued rigorous work in this important field has potential to greatly improve patient outcomes. I have full confidence that with further refinement, this review will make a valuable contribution to advancing precision cancer care for brain metastases.

Reviewer 2 Report

Comments and Suggestions for Authors

ijms-3034698

Type of manuscript: Review

Title: A Review of the Molecular Determinants of Therapeutic Response in Non-Small Cell Lung Cancer Brain Metastases

Authors: Catherine Boldig *, Kimberly Boldig, Sepideh Mokhtari, Arnold B

This paper is a review article on the treatment of NSCLC lung cancer metastasized to the brain. The content is deemed appropriate as such. However, due to numerous technical errors in the paper writing process, it is difficult to read and assess the paper properly. Therefore, it is believed that there is a need to carefully correct the pointed-out issues and conduct a review once again.

[Major concerns]

1.   The citation numbers of the references within the main text are formatted incorrectly. Please correct all of them.

2.   Abbreviation: Numerous abbreviations were utilized in the paper, and it is essential to consolidate these abbreviations separately in the abstract from the main text. In cases where abbreviations are used within figures or tables, please list these abbreviations along with their corresponding full names in the figure legends or at the bottom of corresponding tables. If there are two or more abbreviations, arrange them in alphabetical order.

3.   The English writing format on pages 7 to 10 of the paper is incorrect. Please correct them.

4.   Numbering of Figures and Tables: The table and figure numbers in the IJMS paper use Arabic numerals, so please correct them all.

5.   The notation for genes and proteins in the human body is different. For genes, the name should be written in italics. In this paper, the distinction between genes and proteins is ambiguous.

6.   After the paper has been properly revised as described above, the content will likely be reviewed in the second review. It is very troublesome to review the content of a poorly formatted review paper.

Overall, the manuscript can be considered to publication after minor revision as indicated above.

Comments on the Quality of English Language

ijms-3034698

Type of manuscript: Review

Title: A Review of the Molecular Determinants of Therapeutic Response in Non-Small Cell Lung Cancer Brain Metastases

Authors: Catherine Boldig *, Kimberly Boldig, Sepideh Mokhtari, Arnold B

This paper is a review article on the treatment of NSCLC lung cancer metastasized to the brain. The content is deemed appropriate as such. However, due to numerous technical errors in the paper writing process, it is difficult to read and assess the paper properly. Therefore, it is believed that there is a need to carefully correct the pointed-out issues and conduct a review once again.

[Major concerns]

1.   The citation numbers of the references within the main text are formatted incorrectly. Please correct all of them.

2.   Abbreviation: Numerous abbreviations were utilized in the paper, and it is essential to consolidate these abbreviations separately in the abstract from the main text. In cases where abbreviations are used within figures or tables, please list these abbreviations along with their corresponding full names in the figure legends or at the bottom of corresponding tables. If there are two or more abbreviations, arrange them in alphabetical order.

3.   The English writing format on pages 7 to 10 of the paper is incorrect. Please correct them.

4.   Numbering of Figures and Tables: The table and figure numbers in the IJMS paper use Arabic numerals, so please correct them all.

5.   The notation for genes and proteins in the human body is different. For genes, the name should be written in italics. In this paper, the distinction between genes and proteins is ambiguous.

6.   After the paper has been properly revised as described above, the content will likely be reviewed in the second review. It is very troublesome to review the content of a poorly formatted review paper.

Overall, the manuscript can be considered to publication after minor revision as indicated above.

Round 2

Reviewer 1 Report

Comments and Suggestions for Authors

Dear colleagues, 

1. After carefully reviewing the paper, here are three suggestions to elevate its caliber:

- Strengthen the introduction by providing deeper background on the epidemiology and pathophysiology of lung cancer brain metastases. This will give important context for non-experts.

- Incorporate more recent literature, as some references date back over a decade. Ensuring currency is vital for a comprehensive review. 

- Consider expanding the discussion of moleculartargeted therapies and immunotherapy. Your analysis of precision medicine strategies could be more comprehensive.

2. The language is suitably academic and precise as expected for this peer-reviewed journal. For example, molecular pathways involved in oncogenesis and metastasis are clearly depicted and defined. Some terms, like "oncogenic drivers", are expertly introduced for clarity. Occasional simpler verbs like "occur" could be replaced with more specific medical terminology to align with the highly specialized topic.

3. A key strength is its thorough overview of genetics and targeted therapies for different lung cancer subtypes. However, expanding discussion of differential brain metastasis rates and intracranial response patterns could provide deeper insights. 

4. To the authors, keep pushing boundaries in this important area. Your efforts advance science and help more patients. Persist through challenges, collaborate with others, and stay passionate. Impactful research often stems from teamwork and perseverance over time. Thank you for your contributions.

I hope these perspectiveprove perspective. Please let me know if any part requires elaboration. I wish you the best moving forward.

Comments on the Quality of English Language

Dear colleagues, 

1. After carefully reviewing the paper, here are three suggestions to elevate its caliber:

- Strengthen the introduction by providing deeper background on the epidemiology and pathophysiology of lung cancer brain metastases. This will give important context for non-experts.

- Incorporate more recent literature, as some references date back over a decade. Ensuring currency is vital for a comprehensive review. 

- Consider expanding the discussion of moleculartargeted therapies and immunotherapy. Your analysis of precision medicine strategies could be more comprehensive.

2. The language is suitably academic and precise as expected for this peer-reviewed journal. For example, molecular pathways involved in oncogenesis and metastasis are clearly depicted and defined. Some terms, like "oncogenic drivers", are expertly introduced for clarity. Occasional simpler verbs like "occur" could be replaced with more specific medical terminology to align with the highly specialized topic.

3. A key strength is its thorough overview of genetics and targeted therapies for different lung cancer subtypes. However, expanding discussion of differential brain metastasis rates and intracranial response patterns could provide deeper insights. 

4. To the authors, keep pushing boundaries in this important area. Your efforts advance science and help more patients. Persist through challenges, collaborate with others, and stay passionate. Impactful research often stems from teamwork and perseverance over time. Thank you for your contributions.

I hope these perspectiveprove perspective. Please let me know if any part requires elaboration. I wish you the best moving forward.

Reviewer 2 Report

Comments and Suggestions for Authors

Accept the current revised manuscript.

Author Response

Thank you!